# Geometric Signatures as Important Factors to Control the Photo-Stabilities of the Phosphorescent Pd(II)/Pt(II) Complexes: A Case Study

**DOI:** 10.3390/molecules28124587

**Published:** 2023-06-06

**Authors:** Yafei Luo, Lingkai Tang, Zhongzhu Chen, Zhigang Xu, Yanan An, Mingyao Li, Jianping Hu, Dianyong Tang

**Affiliations:** 1National & Local Joint Engineering Research Center of Targeted and Innovative Therapeutics, Chongqing Engineering Laboratory of Targeted and Innovative Therapeutics, Chongqing Key Laboratory of Kinase Modulators as Innovative Medicine, Chongqing Collaborative Innovation Center of Targeted and Innovative Therapeutics, College of Pharmacy (International Academy of Targeted Therapeutics and Innovation), Chongqing University of Arts and Sciences, Chongqing 402160, China; luoyf@cqwu.edu.cn (Y.L.); 18883138277@163.com (Z.C.); xzg@cqwu.edu.cn (Z.X.); zhiyaoanyanan@163.com (Y.A.); limingyao0524@163.com (M.L.); 2Key Laboratory of Medicinal and Edible Plants Resources Development of Sichuan Education Department, School of Pharmacy, Chengdu University, Chengdu 610106, China; tlk520666@163.com

**Keywords:** operation lifetime, tetradentate transition metal complex, intramolecular interactions, intermolecular interactions, TTA (triplet-triplet annihilation), OLED

## Abstract

Operation lifetime, as an important parameter, determines the performance of phosphorescent organic light-emitting diodes (OLEDs). Unveiling the intrinsic degradation mechanism of emission material is crucial for improving the operation’s lifetime. In this article, the photo-stabilities of tetradentate transition metal complexes, the popular phosphorescent materials, are explored by means of density functional theory (DFT) and time-dependent (TD)-DFT, aiming to illustrate the geometric signatures as important factors to control the photo-stabilities. Results indicate that for the tetradentate Ni(II), Pd(II), and Pt(II) complexes, the coordinate bonds of the Pt(II) complex exhibit stronger strength. It seems that the strengths of coordinate bonds are closely related to the atomic number of the metal center in the same group, which could be attributed to the various electron configurations. The effect of intramolecular and intermolecular interactions on ligand dissociation is also explored here. The large intramolecular steric hindrance and strong π-π interaction between the Pd(II) complexes caused by aggregation could effectively raise the energy barriers of the dissociation reaction, leading to an unfeasible reaction pathway. Moreover, the aggregation of Pd(II) complex can change the photo-deactivation mechanism as compared to that of monomeric Pd(II) complex, which is favored for avoiding the TTA (triplet-triplet annihilation) process.

## 1. Introduction

Pt(II) and Pd(II) complexes have attracted more and more attention because of their excellent virtues. Essentially, the strong spin-orbital coupling (SOC) effect can facilitate the effective intersystem crossing (ISC) process and mixture of singlet and triplet excited states, causing the forbidden T_1_→S_0_ transition to become acceptable. The unique property of excited state transformation can promote the application of Pt(II) and Pd(II) complexes in many hot fields. For instance, Pt(II) and Pd(II) complexes can be regarded as phosphorescent emitters in the field of OLEDs. Over the past decades, substantial progress has been made in the development of high-performance Pt(II) and Pd(II) complexes as a result of extensive research within the academic and industrial communities, which has enabled phosphorescent OLEDs to become a main component in state-of-the-art displays as well as in next-generation solid-state lighting [1,2,3,4,5]. Secondly, as a consequence of their efficient intersystem crossing derived from metal-to-ligand charge transfer (MLCT), Pt(II) and Pd(II) complexes have been developed as popular candidates [6,7]. Recently, Wong and co-workers proposed that well-designed J-aggregates of organometallic complex molecules are an efficient strategy to realize NIR-II phosphorescence-based hypoxia bioimaging [8]. Thirdly, the Pt(II) and Pd(II) complexes can be used as antitumor drugs, which can exhibit cell-killing effects. In terms of antiproliferative activity, it has been revealed that the newly synthesized Pd(II) complex had stronger cytotoxic activity in the two tested human tumor cell lines than the remaining complexes, which suggests its promising potential as an antitumor drug [9]. Zhu et al. have reported Pt(IV) complex as an antitumor prodrug, and it can be controllably activated with the help of red light [10].

These complexes act as important functional components in the phosphorescence-based hypoxia bioimaging, antitumor prodrug, and phosphorescent OLEDs; undoubtedly, the photo-stability is crucial because it determines the performance in practical applications. Herein, we take complex applied as a phosphorescent emitter in the field of OLEDs as an example to illustrate the role of photo-stability. During the OLED’s operation, generally, once it possesses a high brightness range, it would experience a drastic drop in efficiency, that is, efficiency roll-off in phosphorescent emitters, which originates from triplet-triplet annihilation (TTA) and triplet-polaron quenching (TPQ) [11,12]. It is demonstrated that the TTA or TPQ processes in the emissive layer (EML) are closely related to the device’s operational stability because the TTA or TPQ processes can result in the generation of hot (multiply excited) excitons or polarons. The hot (multiply excited) excitons or polarons could induce chemical bond dissociation of the guest or host molecule; subsequently, an expedited device degradation process will occur, especially under conditions of high-brightness operation [13,14]. Thus far, enormous efforts have been made to improve the device lifetime of phosphorescent OLEDs, and some promising strategies have been provided: (I) Developing rigid phosphorescent dopants can improve the stability needed to sustain the excited state. The studies explored by Li and co-workers exhibited that the rigidity of molecules can be effectively promoted by employing a tetradentate core, which should be further investigated to enhance emitter stability [1,15,16]. (II) Developing suitably stable host materials is another strategy for addressing the short lifetime of phosphorescent OLEDs. For example, electron-transporting host materials, including a bulky triphenylsilyl unit, are usually applied to enhance the device’s lifetime [17,18]. 

Apart from the above-mentioned strategies, Kim and co-workers reported recently that the photochemical stability of the high-lying metal-centered triplet state can be effectively enhanced by adding bulky 3,5-di-tert-butyl-phenyl into the *N*-heterocyclic carbene moiety in the Pt(II) complex [19]. More importantly, the bulky substituent could prevent undesirable host-guest interactions to some extent. Both the high-lying metal-centered triplet state and the prevention of undesirable host-guest interactions contribute to a longer device lifetime. Li et al. illustrated that a host-free Pd(II)-based OLED had excellent performance and an ultra-long operational half-life. In their study, they declare that phosphorescent molecular aggregates can be feasible emitter candidates for lighting and display applications [20]. Due to the distinctive d^8^-configuration and square planar geometry, Pt(II) and Pd(II) complexes can have a greater tendency to form π-π-stacking interactions, which is a significant signature of geometry. This aggregation can lead to the obvious change of properties of Pt(II) and Pd(II) complexes, for example, the red shift of emission wavelength, phosphorescent efficiency, and operational stability [21,22,23].

As shown in these studies, it indicates that the photo-stabilities or operation lifetimes are closely related to the geometric signatures and intramolecular and intermolecular interactions of tetradentate Pt(II) and Pd(II) complexes. However, a systematical theoretical investigation of the stabilities of tetradentate Pt(II) and Pd(II) complexes so far is insufficient because they exhibit more rigidities as compared to those of their bi- and tridentate counterparts. In addition, some significant and meaningful questions can be provided in the investigations of the stabilities of Pt(II) and Pd(II) complexes: (1) What is the relationship between structural signatures, including coordinated forms, aggregation, and intramolecular interaction, and the photostability of tetradentate Pt(II) and Pd(II) complexes? (2) What is the role of aggregation in the photostabilities of Pt(II) and Pd(II) complexes? (3) Can the molecular aggregation of complexes tune the photo-deactivation mechanism? For the sake of completely answering these questions, some Pt(II) and Pd(II) complexes shown in Figure 1 are chosen, which can explore the role of “metal-coordinate effect”, “aggregation effect”, and “steric hindrance effect” in the photo-stability. Therein, **Pd-1**, **Pt-2,** and **Pt-3** are reported in Li and Kim’s studies, respectively [19,20]. This study can provide significant and meaningful information for designing stable Pt(II) and Pd(II) complexes used in the fields of phosphorescent probes, prodrugs, and OLEDs.

## 2. Computational Details

In this paper, the geometry optimizations and vibrational frequency calculations of ground states (S_0_), lowest-lying triplet excited states (T_1_), cation states (+), anion states (−), and lowest-lying singlet excited states (S_1_) for all the tetradentate Pt(II) and Pd(II) complexes are presented in the methodology of restricted, unrestricted, and time-dependent DFT (RDFT, UDFT, and TDDFT) with the B3LYP hybrid functional [24], respectively. As shown in the previous calculations of various Pt(II) complexes, the B3LYP hybrid functional can well reproduce the experimental results [25,26,27]. The D3BJ dispersion correction [28] was also taken into account. The mix basis sets, including the LANL2DZ basis set [29] and the 6-311G(d,p) basis set [30,31], were used to describe the heavy atoms Pt/Pd and the light atoms (C, H, O, and N). These calculations can be performed in the Gaussian 16 software [32]. 

In the application of optimized geometry to the T_1_ excited state, the spin-orbit coupling matrix elements were evaluated, which was realized by means of linear (single residue) response theories. The calculated protocol is the TDDFT with the B3LYP functional, as implemented in the Dalton program [33]. Moreover, the quadratic response theories in the methodology of TDDFT were used to obtain the radiative rate constants. During these computations, the LANL2DZ_ECP and 6-31G for Ir and light atoms are used to balance computational costs and accuracies. The other parameters, including topological parameters of the electron density and Laplacian bond orders, were calculated using Multiwfn_3.8 software [34]. The pictures of geometry, spin density, and interaction region indicator (IRI) analysis were drawn by combining the VMD [35] and Multiwfn_3.8 software.

The molecular dynamics (MD) simulation was performed using the GROMACS 2018.8 software [36] with periodic boundary conditions. A box containing one **Pd-1** molecule and 30 mCBP molecules was built by the Packmol program [37]. The NPT ensemble was selected for a duration of 1 ns to obtain a reasonable density at 298 K and 1 bar pressure, followed by a 100 ns NVT production simulation under the same conditions. The GAFF force field [38] was adopted to describe the molecules. The Nosé-Hoover thermostat was used to control the temperature, and the pressure was controlled by the Parrinello-Rahman in an isotropic manner to reach equilibrium. 

## 3. Results and Discussion

### 3.1. The Influence of the “Metal-Coordinate Effect” on the Stability of Tetradentate Complexes

In order to explore the bonding nature between the Ni/Pd/Pt and the ligand, the electron localization function (ELF) was carried out here, and two-dimensional color-filled maps of S_0_ states are depicted in Figure 1. The two-dimensional color-filled maps of T_1_, S_1_, +, and − states for **Ni-1**, **Pd-1,** and **Pt-1** are shown in Appendix A. The two-dimensional color-filled ELF maps can be applied to show the coordination electron density localization. As shown in Figure 1, for all complexes, namely **Ni-1**, **Pd-1,** and **Pt-1**, the ELFs between Ni/Pd/Pt and N atoms are smaller than those between Ni/Pd/Pt and C atoms, indicating that the degree of electron localization between Ni/Pd/Pt-N bonds is smaller than that between Ni/Pd/Pt-C bonds. Moreover, from **Ni-1** to **Pd-1 and Pt-1**, the ELF analysis can suggest that the electron between the Pt, N, and C atoms shows gradually localized features in the S_0_, T_1_, S_1_, +, and − states, respectively. In addition, for the **Ni-1** complex, compared with the T_1_ and S_1_ states, it seems that the ELFs between Ni and ligand are smaller than those in the S_0_, +, and − states. On the basis of ELF analyses, one can speculate that, firstly, the electron-localized feature between N and Ni/Pd/Pt atoms could lead to decreased photo-stability of tetradentate Pt(II) complexes. Secondly, the coordinated bond could possess the tendency that, as the atomic number increases, the strengths should increase in the same group of periodic tables of the elements. Thirdly, for the **Ni-1** complex, the smaller electron localization between N and Ni in T_1_ and S_1_ states as compared to those in S_0_, +, and − states imply that the unstable coordinated bonds may be causing the unsatisfactory photo-stability. 

Next, a topological analysis was performed to investigate the properties of the coordinate bond further. The corresponding results were collected in Appendix A. The parameters of the geometries are shown in Appendix A. The ρ_BCP_ of the Ni/Pd/Pt-N bonds are smaller than the others in the **Ni-1** to **Pd-1** and **Pt-1**, which indicates that the ρ(r) values between Ni/Pd/Pt and N are the smallest. Similarly, the ρ(r) values of all coordinate bonds tend to follow the trend in **Ni-1** < **Pd-1 < Pt-1** as a whole. In the case of **Ni-1**, the ρ(r) values of Ni-C and Ni-N bonds are also smaller than those in other states. These computed results are consistent with those of the ELF analyses. Here, all ∇^2^ρ_BCP_ were larger than 0, therefore indicating electron depletion at the BCP. Furthermore, the E_BCP_ values are almost less than or close to 0, which is sufficient to contribute toward bond stabilization [39,40]. The Laplacian bond orders (LBO) of these coordinate bonds were calculated, and the results are summarized in Table 1. Obviously, the LBO of Ni/Pd/Pt-N bonds are considerably smaller than that of the other Ni/Pd/Pt-C bonds. For **Ni-1**, during the transition from S_0_ to T_1_ and S_1_, the Ni-N bonds would further weaken. These results demonstrated that the Ni/Pd/Pt-N bonds are the weakest bonds among all complexes, and the sequence of photostabilities may be as follows: **Ni-1** < **Pd-1** < **Pt-1**.

Finally, the reaction pathways of the Ni/Pd/Pt-N bond breaking were computed to intuitively illustrate the photo-stabilities of **Ni-1**, **Pd-1,** and **Pt-1**, respectively. The corresponding vibration forms of TS are shown in Appendix A. The calculated reaction steps are pictured in Figure 2. For the **Ni-1** complex, the energy barriers are 27.06, 18.96, 16.61, 28.35, and 29.52 kcal/mol in the S_0_, T_1_, S_1_, +, and − states, respectively. Seen from these energy barriers, compared to the S_0_, +, and − states, the **Ni-1** complex could exhibit weak photo-stability in the T_1_ and S_1_ states, which are the predominant states involved in the phosphorescent process. For the **Pd-1** complex, the Pd-N bond breaking could need to overcome the energy barriers of 23.88, 22.71, 23.15, 27.80, and 24.11 kcal/mol, indicating that the stability of the **Pd-1** complex, in comparison, is stronger than that of the **Ni-1** complex. In the investigation of stability for **Pt-1**, larger energy barriers can be needed as compared to those of **Ni-1** and **Pd-1,** and the corresponding values are 30.56, 31.76, 31.68, 35.06, and 30.56 in various states. Analyses of the energy barriers can indicate that, from a structural viewpoint, the stabilities of transition-metal complexes seem to be closely related to the atomic numbers of metal centers in the same group. Additionally, the calculated energy barriers agree with the conclusions of ELF, topological analysis, and Laplacian bond order. 

### 3.2. The Effect of Steric Hindrance on the Stability of Tetradentate Metal Complexes

Here, the effect of intramolecular steric hindrance on the stability of tetradentate Pt(II) complexes, namely **Pt-2** and **Pt-3**, is explored. In order to explore the bonding nature between the Pt and ligand, the ELF, topological analysis of the electron density, and Laplacian bond orders were first calculated, and the results of S_0_ states are shown in Figure 3, Appendix A. The parameters of the geometries are shown in Appendix A. As polled in Figure 3, for **Pt-2** and **Pt-3**, the ELFs between Pt, N, and C atoms are similar in different states, implying that the degree of electron localization between Pt-N and Pt-C atom bonds is comparable. Compared with the Pt-C bonds, a smaller electron localization between Pt and N can be found. The ρ_BCP_ and Laplacian bond orders of the Pt-C and N bonds are also applied to demonstrate the reliability of EFL analysis. On the basis of the results of ELF and topological analysis of the electron density and Laplacian bond orders, it can be indicated that the strength of Pt-N bonds, is weaker than that of Pt-C bonds and the substituent can have a negligible influence on the coordinate bonds. 

As shown in the results of ELF, topological analysis of the electron density and Laplacian bond orders revealed the reaction pathways of the Pt-N bond breaking for the **Pt-2** and **Pt-3** at different states, which are shown in Figure 4. The corresponding vibration forms of TS are shown in Appendix A. For **Pt-2**, the steps of Pt-N bond breaking would overcome the energy barriers of 22.77, 17.31, 24.56, and 18.80 kcal/mol S_0_, T_1_, +, and − states, respectively. In comparison to **Pt-2**, larger energy barriers can be achieved, and the corresponding values are 23.94, 19.17, 25.51, and 23.94 kcal/mol, respectively. Although the relative difference in energy barrier is not large enough, the tendency of energy barrier can illustrate that the Pt-N bond breaking of **Pt-3** is more difficult than that of **Pt-2**. According to the experimental results [19], introducing bulky substituents in the Pt(II) complex can enhance the photochemical stability; hence, the calculated results are consistent with the experimental ones. To unveil the role of 3,5-di-tert-butyl-phenyl, only the isosurface map of the IRI of S_0_ states was computed because of the similar geometries in excited states. Figure 5 shows the isosurface map of the IRI of S_0_ states for **Pt-2** and **Pt-3** complexes, and *sign(λ2)ρ* is mapped on the isosurfaces according to the coloring method. The IRI analysis indicates that for the S_0_ state of **Pt-2**, there is a small interaction. Nevertheless, compared with **Pt-2**, a large intramolecular steric hindrance appears in the S_0_ states of **Pt-3**, which originates from the 3,5-di-tert-butyl-phenyl group. 

### 3.3. The Effect of the “Aggregation Effect” on the Stability of Tetradentate Metal Complexes

The ELF, topological analysis of the electron density, and Laplacian bond orders were first calculated to illustrate the bonding nature between the Pd and ligand for the **Pd-1D** and **Pd-1T** at S_0_ and T_1_ states, and the results are shown in Appendix A. As polled in Appendix A, for **Pt-2** and **Pt-3**, the ELFs between Pt, N, and C atoms are similar in different states, implying that the degree of electron localization between Pt-N and Pt-C atomic bonds is comparable. Compared with the Pt-C bonds, a smaller electron localization between Pt and N can be found. The ρBCP and Laplacian bond orders of the Pt-C and N bonds are also applied to demonstrate the reliability of EFL analysis. On the basis of the results of ELF and topological analysis of the electron density and Laplacian bond orders, it can be indicated that the strength of Pt-N bonds is weaker than that of Pt-C bonds, and the aggregation can have a negligible influence on the coordinate bonds.

For the sake of exhibiting the “aggregation effect” on the stability of tetradentate metal complexes, the reaction pathways of the Pt-N bond breaking of **Pd-1**, **Pd-1D,** and **Pd-1T** are investigated with the help of a rigidity scan. Therein, the **Pd-1** is trapped in the host materials, and the molecular dynamics (MD) simulation was performed using the GROMACS 2018.8 software. The corresponding results of rigidity scans for **Pd-1**, **Pd-1D,** and **Pd-1T** are shown in Figure 6. As plotted in Figure 6, the energy barriers of **Pd-1D** and **Pd-1T** in S_0_ and T_1_ states are far lager than those of **Pd-1**, indicating that the “aggregation effect” can cause the excellent stability of tetradentate metal complexes. This result is very much in accordance with the experimental investigation. In addition, the large energy barriers pictured in Figure 6 indicate that Pt-N bond breaking is almost impossible at room temperature.

The isosurface map of IRI states for **Pd-1**, **Pd-1D,** and **Pd-1T** complexes, based on the MD snapshot and geometric optimization, was also carried out to illustrate the intermolecular interaction. The corresponding results were plotted in Figure 7. As exhibited in Figure 7, for the host-guest system, the intermolecular interaction between **Pd-1** and mCBP can be classified as π-π interaction and other weak interactions. Compared with the **Pd-1** in the doped condition, the isosurface maps of IRI for **Pd-1D** and **Pd-1T** in Figure 7 exhibit that there is strong π-π interaction between the Pd(II) complexes. As shown in the **Pd-1T** complex, the π-π interaction and the middle Pd(II) complex can construct the “sandwich structure” to form the restricted region. This restricted region can effectively constrain the geometric change of the **Pd-1T** complex, which leads to a high energy barrier for the **Pd-1T**. In other words, the ligand is restricted in the special region to form the coordinate bonds with Pd, and the dissociation of complexes could originate from the bonds of the ligand, which may need a high energy barrier due to the property of covalent bonds.

### 3.4. Investigation of the Influence of Transformation between Singlet and Triplet Excited States on Stability

Apart from the investigations of the influence of intramolecular/intermolecular interactions and intrinsic properties on stability, the transformation between singlet and triplet excitons is also considered to explore the effect factor of stability. The calculations of adiabatic singlet and triplet excited states were carried out with the help of Tamm-Dancoff approximation density functional theory (TDA-DFT) in the protocol of B3LYP with LANL2DZ for Pd and6-311G** for C, H, N, and O. The calculated T_1_ energies of **Pd-1** and **Pd-1D** are 2.55 eV and 1.98 eV, respectively. In order to further understand the intersystem crossing processes of **Pd-1** and **Pd-1D**, the exploration of the energy-level alignment of singlet and triplet excited states is very significant since the transformation mechanism from singlet to triplet excited states may depend on the energy-level alignment. Hence, the S_1_ and T_2_ energy levels of **Pd-1** and **Pd-1D** are further computed. As shown in Figure 8, the calculated S_1_/T_2_ energies of **Pd-1** and **Pd-1D** in gas phase are calculated to be 2.81 eV/2.73 eV and 2.08 eV/2.33, respectively. It is interesting to note that the S_1_ energy of **Pd-1** is higher than the T_2_ energy. In comparison, the calculated T_2_ energy of **Pd-1D** is higher than the S_1_ energy. For the calculated energy-level alignment of **Pd-1**, due to the competitive spin conversion process (S_1_ → T_2_ and S_1_ → T_1_), the intersystem crossing (ISC) may be inefficient. On the contrary, in the case of **Pd-1D**, because of the endothermic energy-level alignment, the ISC is only from S_1_ to T_1_, not from S_1_ to T_2_.

To explore the transformation between singlet and triplet excited states in detail, the intersystem crossing (ISC) rates of **Pd-1** (S_1_ → T_2_ and S_1_ → T_1_) and **Pd-1D** (S_1_ → T_1_) are calculated, which is plotted in Figure 7. The computed intersystem crossing (ISC) rates of **Pd-1** (S_1_ → T_2_ and S_1_ → T_1_) are 1.00 × 10^10^ s^−1^ and 7.44 × 10^9^ s^−1^, respectively. The computed results indicate that the intersystem crossing (ISC) from S_1_ to T_2_ is prior to that of S_1_ → T_1_. For the **Pd-1D**, the calculated intersystem crossing (ISC) from S_1_ to T_1_ is 5.45 × 10^9^ s^−1^. The radiative decay rates for **Pd-1** (T_2_ → S_0_ and T_1_ → S_0_) and **Pd-1D** (T_1_ → S_0_) are also computed, and the corresponding values are 4.84 × 10^2^ s^−1^, 3.79 × 10^4^ s^−1^, and 4.38 × 10^4^ s^−1^, respectively. The computed radiative decay rates indicate that for **Pd-1**, the phosphorescence originated from the T_1_ excited state. In addition, compared with the **Pd-1**, the **Pd-1D** possesses a larger radiative decay rate, which is beneficial for facilitating the transition of the triplet exciton to the S_0_ state. 

On the basis of the above-mentioned discussion, the phosphorescent process of **Pd-1D** could be identified as S_1_ → T_1_ → S_0_. However, in the case of the **Pd-1** complex, the corresponding process is S_1_ → T_2_ →T_1_→ S_0_. In comparison to the **Pd-1D** complex, the complicated photo-physical process of **Pd-1** can cause aggregation of triplet density in the emission layer (EML). The long excited state lifetime of triplet excitons is favored for resulting in a triplet-triplet quenching process. Triplet-triplet annihilation (TTA) and triplet-polaron annihilation (TPA) are the main degradation mechanisms. Surely, this conclusion needs to be confirmed via more investigation, and herein, only thinking is put forward in our study.

After exploration of the effect of the metal-N coordinate on the stability, the ligand dissociations of **Ni-1**, **Pd-1**, **Pt-1**, **Pt-2,** and **Pt-3** are also investigated. As shown in Appendix A, the bond dissociation energy of **Pd-1** is smaller than that of **Ni-1**. Among the **Ni-1**, **Pd-1**, and **Pt-1**, the **Pt-1** possesses the largest bond dissociation energies in the S_0_, T_1_, S_1_, +, and − states, indicating that the different coordinate circumstances can induce the different stabilities. In the case of **Pt-2** and **Pt-3**, the bond dissociation energies in the S_0_, T_1_, S_1_, +, and − states are comparable, implying that the substituent can cause a slight effect on the stability. In fact, for **Ni-1**, **Pd-1**, **Pt-1**, **Pt-2,** and **Pt-3**, the large bond dissociation energies could largely suppress the ligand dissociation. Hence, the investigation of breakage of the metal-N coordinate may be a reasonable strategy to unveil the stability of the tetradentate transition metal complex.

Finally, the dipole moment is also simulated. As shown in the previous study [41], the orientation of metal complexes is related to the dipole moment, and a molecular-scale simulation of the film formation suggested that dipole-dipole interactions tend to hinder alignment in organic films [42,43]. The dipole moments are computed here for **Ni-1**, **Pd-1**, **Pt-1**, **Pt-2,** and **Pt-3,** and the corresponding results are shown in Appendix A. For **Ni-1**, **Pd-1,** and **Pt-1**, the simulated dipole moment vectors are nearly identical, implying that the various mental centers can result in an ignorable effect. In the case of **Pt-2**, the dipole moment vector is parallel to the molecule. In comparison, for **Pt-3**, the dipole moment vector is pulled out of the molecular plane. On the basis of the simulated results, it can be inferred that the substituent added to the ligand can cause a distinct change in the dipole moment vector. In some situations, the dipole-dipole interactions do not play a significant role in the orientation of transition metal complexes [44]. Without doubt, the orientation of metal complexes in amorphous films is very complicated and should be explored using more methods.

## 4. Conclusions

In this investigation, a systematic investigation of the degradation mechanisms of tetradentate Pt(II) and Pd(II) complexes has been performed to illustrate the geometric signatures as important factors to control the chemical stabilities. The corresponding conclusions can be put forward as follows:(I).Compared with the tetradentate Ni(II) and Pd(II) complexes, the Pt(II) complex possesses more strong coordinate bonds, which suggests that the strengths of coordinate bonds are closely related to the atomic number of the metal center in the same group.(II).The large intramolecular steric hindrance and strong π-π interaction between the complexes originated from aggregation can efficiently constrain the geometric change, raising the energy barriers of the ligand dissociation reaction. (III).In comparison to the monomeric Pd(II) complex, the aggregation of the Pd(II) complex can result in the essential change of the photo-deactivate mechanism and reduce the lifetime of triplet excitability, which is beneficial for avoiding triplet-triplet annihilation.

According to these conclusions, the effect factors of tetradentate metal complexes have been explored in detail. This study can provide meaningful strategies and information for designing high-stability transition metal complexes as phosphorescent materials.

## Data Availability

Not applicable.

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
