# Peer review of "Geometric Signatures as Important Factors to Control the Photo-Stabilities of the Phosphorescent Pd(II)/Pt(II) Complexes: A Case Study"

_molecules, 2023, doi:10.3390/molecules28124587_

Round 1

Reviewer 1 Report

The author investigated the degradation mechanism of tetradentate Pt(II) and Pd(II) complexes have been performed to illustrate the geometric signatures as important factors in controlling the chemical stabilities. The overall work is good and can be published after major revision by addressing the following comments.

1.     Why did the authors choose Ni, Pd, and Pt Complexes only? Because Ir complexes also show chemical stability near Pt.

2.     In scheme 1, the aggregation effect is in the second position, but the results and discussion are different flows.

3.     Bond dissociation energy is also the main key for stability. So providing such information also supports the work.

4.     Authors can include the Ir complex also, which is widely used in phosphorescent OLEDs.

5.    Simulated dipole moment vector should be provided for understanding the orientation factor. 

Reviewer 2 Report

The article reports the detailed analysis of some Ni(II), Pd(II), and Pt(II) complexes which are promising for OLED applications. The article is interesting and brings novel analysis as to the photo physical properties of the systems. In view of the reported data I recommend the article for publishing after minor corrections. 

The methodology section should be extended. It is not clear how Authors computed intersystem crossing rates. 

The article is well written, but requires language correction.  

Round 2

Reviewer 1 Report

The manuscript is improved and can be accepted in its present form.